# Transportability from Multiple Environments with Limited Experiments

**Elias Bareinboim**[*]
UCLA

**Sanghack Lee**[*]
Penn State University

**Vasant Honavar**
Penn State University

**Judea Pearl**
UCLA

## Abstract

This paper considers the problem of transferring experimental findings learned from multiple heterogeneous domains to a target domain, in which only limited experiments can be performed. We reduce questions of transportability from multiple domains and with limited scope to symbolic derivations in the causal calculus, thus extending the original setting of transportability introduced in [1], which assumes only one domain with full experimental information available. We further provide different graphical and algorithmic conditions for computing the transport formula in this setting, that is, a way of fusing the observational and experimental information scattered throughout different domains to synthesize a consistent estimate of the desired effects in the target domain. We also consider the issue of minimizing the variance of the produced estimand in order to increase power.

## 1   Motivation

Transporting and synthesizing experimental knowledge from heterogeneous settings are central to scientific discovery. Conclusions that are obtained in a laboratory setting are transported and applied elsewhere in an environment that differs in many aspects from that of the laboratory. In data-driven sciences, experiments are conducted on disparate domains, but the intention is almost invariably to fuse the acquired knowledge, and translate it into some meaningful claim about a target domain, which is usually different than any of the individual study domains.

However, the conditions under which this extrapolation can be legitimized have not been formally articulated until very recently. Although the problem has been discussed in many areas of statistics, economics, and the health sciences, under rubrics such as "external validity" [2, 3], "meta-analysis" [4], "quasi-experiments" [5], "heterogeneity" [6], these discussions are limited to verbal narratives in the form of heuristic guidelines for experimental researchers – no formal treatment of the problem has been attempted to answer the practical challenge of generalizing causal knowledge across multiple heterogeneous domains with disparate experimental data posed in this paper.

The fields of artificial intelligence and statistics provide the theoretical underpinnings necessary for tackling transportability. First, the distinction between statistical and causal knowledge has received syntactic representation through causal diagrams [7, 8, 9], which became a popular tool for causal inference in data-driven fields. Second, the inferential machinery provided by the causal calculus (do-calculus) [7, 9, 10] is particularly suitable for handling knowledge transfer across domains.

Armed with these techniques, [1] introduced a formal language for encoding differences and commonalities between domains accompanied with necessary or sufficient conditions under which transportability of empirical findings is feasible between two domains, a source and a target; then, these conditions were extended for a complete characterization for transportability in one domain with unrestricted experimental data [11]. Subsequently, these results were generalized for the settings when

---

[*]These authors contributed equally to this paper. The authors' addresses are respectively eb@cs.ucla.edu, sxl439@ist.psu.edu, vhonavar@ist.psu.edu, judea@cs.ucla.edu.

only limited experiments are available in the source domain [12, 13], and further for when multiple source domains with unrestricted experimental information are available [14, 15]. This paper broadens these discussions introducing a more general setting in which multiple heterogeneous sources with limited and distinct experiments are available, a task that we call here "$mz$-transportability".[1]

More formally, the $mz$-transportability problem concerns with the transfer of causal knowledge from a heterogeneous collection of source domains $\Pi = \{\pi_1, ..., \pi_n\}$ to a target domain $\pi^*$. In each domain $\pi_i \in \Pi$, experiments over a set of variables $\mathbf{Z}_i$ can be performed, and causal knowledge gathered. In $\pi^*$, potentially different from $\pi_i$, only passive observations can be collected (this constraint is weakened later on). The problem is to infer a causal relationship $R$ in $\pi^*$ using knowledge obtained in $\Pi$. Clearly, if nothing is known about the relationship between $\Pi$ and $\pi^*$, the problem is trivial; no transfer can be justified. Yet the fact that all scientific experiments are conducted with the intent of being used elsewhere (e.g., outside the lab) implies that scientific progress relies on the assumption that certain domains share common characteristics and that, owed to these commonalities, causal claims would be valid in new settings even where experiments cannot be conducted.

The problem stated in this paper generalizes the one-dimensional version of transportability with limited scope and the multiple dimensional with unlimited scope. Remarkably, while the effects of interest might not be individually transportable to the target domain from the experiments in any of the available sources, combining different pieces from the various sources may enable the estimation of the desired effects (to be shown later on). The goal of this paper is to formally understand under which conditions the target quantity is (non-parametrically) estimable from the available data.

## 2 Previous work and our contributions

Consider Fig. 1(a) in which the node $S$ represents factors that produce differences between source and target populations. Assume that we conduct a randomized trial in Los Angeles (LA) and estimate the causal effect of treatment $X$ on outcome $Y$ for every age group $Z = z$, denoted by $P(y|do(x), z)$. We now wish to generalize the results to the population of the United States (U.S.), but we find the distribution $P(x, y, z)$ in LA to be different from the one in the U.S. (call the latter $P^*(x, y, z)$). In particular, the average age in the U.S. is significantly higher than that in LA. How are we to estimate the causal effect of $X$ on $Y$ in U.S., denoted $R = P^*(y|do(x))$?[2][3]

The selection diagram for this example (Fig. 1(a)) conveys the assumption that the *only* difference between the two populations are factors determining age distributions, shown as $S \rightarrow Z$, while age-specific effects $P^*(y|do(x), Z = z)$ are invariant across populations. Difference-generating factors are represented by a special set of variables called *selection variables* $S$ (or simply $S$-variables), which are graphically depicted as square nodes (■). From this assumption, the overall causal effect in the U.S. can be derived as follows:

$$
\begin{aligned}
R &= \sum_z P^*(y|do(x), z)P^*(z) \\
&= \sum_z P(y|do(x), z)P^*(z)
\end{aligned}
\tag{1}
$$

The last line is the *transport formula* for $R$. It combines experimental results obtained in LA, $P(y|do(x), z)$, with observational aspects of the U.S. population, $P^*(z)$, to obtain an experimental claim $P^*(y|do(x))$ about the U.S.. In this trivial example, the transport formula amounts to a simple re-calibration (or re-weighting) of the age-specific effects to account for the new age distribution. In general, however, a more involved mixture of experimental and observational findings would be necessary to obtain a bias-free estimate of the target relation $R$. Fig. 1(b) depicts the smallest example in which transportability is not feasible even when experiments over $X$ in $\pi$ are available.

In real world applications, it may happen that certain controlled experiments cannot be conducted in the source environment (for financial, ethical, or technical reasons), so only a limited amount

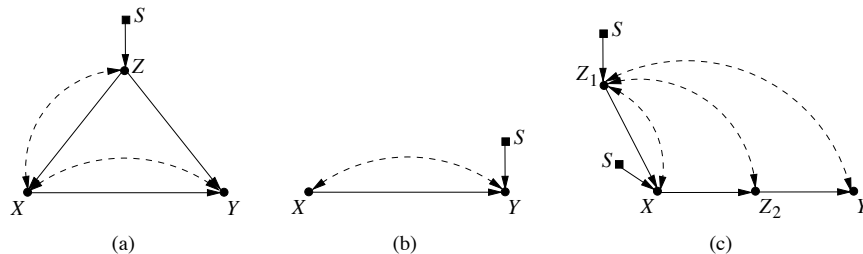

Figure 1: The selection variables $S$ are depicted as square nodes (■). (a) Selection diagram illustrating when transportability between two domains is trivially solved through simple recalibration. (b) The smallest possible selection diagram in which a causal relation is not transportable. (c) Selection diagram illustrating transportability when only experiments over $\{Z_1\}$ are available in the source.

of experimental information can be gathered. A natural question arises whether the investigator in possession of a limited set of experiments would still be able to estimate the desired effects at the target domain. For instance, we assume in Fig. 1(c) that experiments over $Z_1$ are available and the target quantity is $R = P^*(y|do(x))$, which can be shown to be equivalent to $P(y|x, do(Z_1))$, the conditional distribution of $Y$ given $X$ in the experimental study when $Z_1$ is randomized. [4]

One might surmise that multiple pairwise $z$-transportability would be sufficient to solve the $mz$-transportability problem, but this is not the case. To witness, consider Fig. 2(a,b) which concerns the transport of experimental results from two sources ($\{\pi_a, \pi_b\}$) to infer the effect of $X$ on $Y$ in $\pi^*$, $R = P^*(y|do(x))$. In these diagrams, $X$ may represent the treatment (e.g., cholesterol level), $Z_1$ represents a pre-treatment variable (e.g., diet), $Z_2$ represents an intermediate variable (e.g., biomarker), and $Y$ represents the outcome (e.g., heart failure). We assume that experimental studies randomizing $\{Z_1, Z_2\}$ can be conducted in both domains. A simple analysis based on [12] can show that $R$ cannot be $z$-transported from either source alone, but it turns out that combining in a special way experiments from both sources allows one to determine the effect in the target.

More interestingly, we consider the more stringent scenario where only certain experiments can be performed in each of the domains. For instance, assume that it is only possible to conduct experiments over $\{Z_2\}$ in $\pi_a$ and over $\{Z_1\}$ in $\pi_b$. Obviously, $R$ will not be $z$-transported individually from these domains, but it turns out that taking both sets of experiments into account, $R = \sum_{z_2} P^{(a)}(y|do(z_2))P^{(b)}(z_2|x, do(Z_1))$, which fully uses all pieces of experimental data available. In other words, we were able to decompose $R$ into subrelations such that each one is separately $z$-transportable from the source domains, and so is the desired target quantity. Interestingly, it is the case in this example that if the domains in which experiments were conducted were reversed (i.e., $\{Z_1\}$ randomized in $\pi_a$, $\{Z_2\}$ in $\pi_b$), it will not be possible to transport $R$ by any method – the target relation is simply not computable from the available data (formally shown later on).

This illustrates some of the subtle issues $mz$-transportability entails, which cannot be immediately cast in terms of previous instances of the transportability class. In the sequel, we try to better understand some of these issues, and we develop sufficient or (specific) necessary conditions for deciding special transportability for arbitrary collection of selection diagrams and set of experiments. We further construct an algorithm for deciding $mz$-transportability of joint causal effects and returning the correct transport formula whenever this is possible. We also consider issues relative to the variance of the estimand aiming for improving sample efficiency and increasing statistical power.

## 3 Graphical conditions for $mz$-transportability

The basic semantic framework in our analysis rests on *structural causal models* as defined in [9, pp. 205], also called data-generating models. In the structural causal framework [9, Ch. 7], actions are modifications of functional relationships, and each action $do(\mathbf{x})$ on a causal model $M$ produces

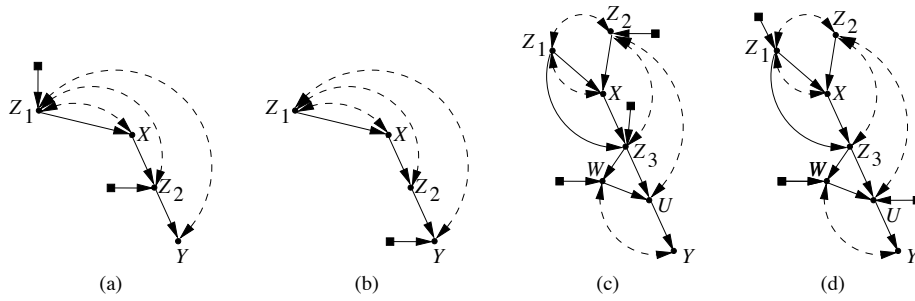

Figure 2: Selection diagrams illustrating impossibility of estimating $R = P^*(y|do(x))$ through individual transportability from $\pi_a$ and $\pi_b$ even when $\mathbf{Z} = \{Z_1, Z_2\}$ (for $(a, b)$, $(c, d)$)). If we assume, more stringently, availability of experiments $\mathbf{Z_a} = \{Z_2\}$, $\mathbf{Z_b} = \{Z_1\}$, $\mathbf{Z}^* = \{\}$, a more elaborated analysis can show that $R$ can be estimated combining different pieces from both domains.

a new model $M_\mathbf{x} = \langle \mathbf{U}, \mathbf{V}, \mathbf{F}_\mathbf{x}, P(\mathbf{U}) \rangle$, where $F_\mathbf{x}$ is obtained after replacing $f_X \in \mathbf{F}$ for every $X \in \mathbf{X}$ with a new function that outputs a constant value $x$ given by $do(\mathbf{x})$. [5]

We follow the conventions given in [9]. We denote variables by capital letters and their realized values by small letters. Similarly, sets of variables will be denoted by bold capital letters, sets of realized values by bold small letters. We use the typical graph-theoretic terminology with the corresponding abbreviations $Pa(\mathbf{Y})_G$ and $An(\mathbf{Y})_G$, which will denote respectively the set of observable parents and ancestors of the node set $\mathbf{Y}$ in $G$. A graph $G_\mathbf{Y}$ will denote the induced subgraph $G$ containing nodes in $\mathbf{Y}$ and all arrows between such nodes. Finally, $G_{\overline{\mathbf{X}}\underline{\mathbf{Z}}}$ stands for the edge subgraph of $G$ where all incoming arrows into $\mathbf{X}$ and all outgoing arrows from $\mathbf{Z}$ are removed.

Key to the analysis of transportability is the notion of "identifiability," defined below, which expresses the requirement that causal effects are computable from a combination of data $P$ and assumptions embodied in a causal graph $G$.

**Definition 1** (Causal Effects Identifiability (Pearl, 2000, pp. 77)). *The causal effect of an action $do(\mathbf{x})$ on a set of variables $\mathbf{Y}$ such that $\mathbf{Y} \cap \mathbf{X} = \emptyset$ is said to be identifiable from $P$ in $G$ if $P_\mathbf{x}(\mathbf{y})$ is uniquely computable from $P(\mathbf{V})$ in any model that induces $G$.*

Causal models and their induced graphs are usually associated with one particular domain (also called setting, study, population, or environment). In ordinary transportability, this representation was extended to capture properties of two domains simultaneously. This is possible if we assume that the structural equations share the same set of arguments, though the functional forms of the equations may vary arbitrarily [11]. [6]

**Definition 2** (Selection Diagram). *Let $\langle M, M^* \rangle$ be a pair of structural causal models [9, pp. 205] relative to domains $\langle \pi, \pi^* \rangle$, sharing a causal diagram $G$. $\langle M, M^* \rangle$ is said to induce a selection diagram $D$ if $D$ is constructed as follows:*

1. *Every edge in $G$ is also an edge in $D$;*

2. *$D$ contains an extra edge $S_i \rightarrow V_i$ whenever there might exist a discrepancy $f_i \neq f_i^*$ or $P(U_i) \neq P^*(U_i)$ between $M$ and $M^*$.*

In words, the $S$-variables locate the *mechanisms* where structural discrepancies between the two domains are suspected to take place. [7] Alternatively, the absence of a selection node pointing to a variable represents the assumption that the mechanism responsible for assigning value to that variable is identical in both domains.

Armed with the concept of identifiability and selection diagrams, $mz$-transportability of causal effects can be defined as follows:

**Definition 3** ($mz$-Transportability). *Let $\mathcal{D} = \{D^{(1)}, ..., D^{(n)}\}$ be a collection of selection diagrams relative to source domains $\Pi = \{\pi_1, ..., \pi_n\}$, and target domain $\pi^*$, respectively, and $\mathbf{Z}_i$ (and $\mathbf{Z}^*$) be the variables in which experiments can be conducted in domain $\pi_i$ (and $\pi^*$). Let $\langle P^i, I_z^i \rangle$ be the pair of observational and interventional distributions of $\pi_i$, where $I_z^i = \bigcup_{\mathbf{Z}' \subseteq \mathbf{Z}_i} P^i(\mathbf{v}|do(\mathbf{z}'))$, and in an analogous manner, $\langle P^*, I_z^* \rangle$ be the observational and interventional distributions of $\pi^*$. The causal effect $R = P_{\mathbf{x}}^*(\mathbf{y}|\mathbf{w})$ is said to be $mz$-transportable from $\Pi$ to $\pi^*$ in $\mathcal{D}$ if $P_{\mathbf{x}}^*(\mathbf{y}|\mathbf{w})$ is uniquely computable from $\bigcup_{i=1,...,n} \langle P^i, I_z^i \rangle \cup \langle P^*, I_z^* \rangle$ in any model that induces $\mathcal{D}$.*

The requirement that $R$ is uniquely computable from $\langle P^*, I_z^* \rangle$ and $\langle P^i, I_z^i \rangle$ from all sources has a syntactic image in the causal calculus, which is captured by the following sufficient condition.

**Theorem 1.** *Let $\mathcal{D} = \{D^{(1)}, ..., D^{(n)}\}$ be a collection of selection diagrams relative to source domains $\Pi = \{\pi_1, ..., \pi_n\}$, and target domain $\pi^*$, respectively, and $\mathbf{S_i}$ represents the collection of $S$-variables in the selection diagram $D^{(i)}$. Let $\{\langle P^i, I_z^i \rangle\}$ and $\langle P^*, I_z^* \rangle$ be respectively the pairs of observational and interventional distributions in the sources $\Pi$ and target $\pi^*$. The relation $R = P^*(\mathbf{y}|do(\mathbf{x}), \mathbf{w})$ is $mz$-transportable from $\Pi$ to $\pi^*$ in $\mathcal{D}$ if the expression $P(\mathbf{y}|do(\mathbf{x}), \mathbf{w}, \mathbf{S_1}, ..., \mathbf{S_n})$ is reducible, using the rules of the causal calculus, to an expression in which (1) do-operators that apply to subsets of $I_z^i$ have no $\mathbf{S_i}$-variables or (2) do-operators apply only to subsets of $I_z^*$.*

This result provides a powerful way to syntactically establish $mz$-transportability, but it is not immediately obvious whether a sequence of applications of the rules of causal calculus that achieves the reduction required by the theorem exists, and even if such sequence exists, it is not obvious how to obtain it. For concreteness, we illustrate this result using the selection diagrams in Fig. 2(a,b).

**Corollary 1.** $P^*(y|do(x))$ *is $mz$-transportable in Fig. 2(a,b) with $\mathbf{Z_a} = \{Z_2\}$ and $\mathbf{Z_b} = \{Z_1\}$.*

*Proof.* The goal is to show that $R = P^*(y|do(x))$ is $mz$-transportable from $\{\pi_a, \pi_b\}$ to $\pi^*$ using experiments conducted over $\{Z_2\}$ in $\pi_a$ and $\{Z_1\}$ in $\pi_b$. Note that naively trying to transport $R$ from each of the domains individually is not possible, but $R$ can be decomposed as follows:

$$P^*(y|do(x)) = P^*(y|do(x), do(Z_1)) \tag{2}$$

$$= \sum_{z_2} P^*(y|do(x), do(Z_1), z_2) P^*(z_2|do(x), do(Z_1)) \tag{3}$$

$$= \sum_{z_2} P^*(y|do(x), do(Z_1), do(z_2)) P^*(z_2|do(x), do(Z_1)), \tag{4}$$

where Eq. (2) follows by rule 3 of the causal calculus since $(Z_1 \perp\!\!\!\perp Y|X)_{D_{\overline{X,Z_1}}}$ holds, we condition on $Z_2$ in Eq. (3), and Eq. (4) follows by rule 2 of the causal calculus since $(Z_2 \perp\!\!\!\perp Y|X, Z_1)_{D_{\overline{X,Z_1},\underline{Z_2}}}$, where $D$ is the diagram in $\pi^*$ (despite the location of the $S$-nodes).

Now we can rewrite the first term of Eq. (4) as indicated by the Theorem (and suggested by Def. 2):

$$P^*(y|do(x), do(Z_1), do(z_2)) = P(y|do(x), do(Z_1), do(z_2), \mathbf{S_a}, \mathbf{S_b}) \tag{5}$$

$$= P(y|do(x), do(Z_1), do(z_2), \mathbf{S_b}) \tag{6}$$

$$= P(y|do(z_2), \mathbf{S_b}) \tag{7}$$

$$= P^{(a)}(y|do(z_2)), \tag{8}$$

where Eq. (5) follows from the theorem (and the definition of selection diagram), Eq. (6) follows from rule 1 of the causal calculus since $(\mathbf{S_a} \perp\!\!\!\perp Y|Z_1, Z_2, X)_{D^{(a)}_{\overline{Z_1, Z_2, X}}}$, Eq. (7) follows from rule 3 of the causal calculus since $(Z_1, X \perp\!\!\!\perp Y|Z_2)_{D^{(a)}_{\overline{Z_1, Z_2, X}}}$. Note that this equation matches with the syntactic goal of Theorem 1 since we have precisely $do(z_2)$ separated from $\mathbf{S_a}$ (and $Z_2 \in I_z^a$); so, we can rewrite the expression which results in Eq. (8) by the definition of selection diagram.

Finally, we can rewrite the second term of Eq. (4) as follows:

$$P^*(z_2|do(x), do(Z_1)) = P(z_2|do(x), do(Z_1), \mathbf{S_a}, \mathbf{S_b}) \tag{9}$$

$$= P(z_2|do(x), do(Z_1), \mathbf{S_a}) \tag{10}$$

$$= P(z_2|x, do(Z_1), \mathbf{S_a}) \tag{11}$$

$$= P^{(b)}(z_2|x, do(Z_1)), \tag{12}$$

where Eq. (9) follows from the theorem (and the definition of selection diagram), Eq. (10) follows from rule 1 of the causal calculus since $(\mathbf{S_b} \perp\!\!\!\perp Z_2 | Z_1, X)_{D^{(b)}_{\overline{Z_1}, X}}$, Eq. (11) follows from rule 2 of the causal calculus since $(X \perp\!\!\!\perp Z_2 | Z_1)_{D^{(b)}_{\overline{Z_1}\underline{X}}}$. Note that this equation matches the condition of the theorem, separate $do(Z_1)$ from $\mathbf{S_b}$ (i.e., experiments over $Z_1$ can be used since they are available in $\pi_b$), so we can rewrite Eq. (12) using the definition of selection diagrams, the corollary follows. $\quad\square$

The next condition for $mz$-transportability is more visible than Theorem 1 (albeit weaker), which also demonstrates the challenge of relating $mz$-transportability to other types of transportability.

**Corollary 2.** $R = P^*(\mathbf{y}|do(\mathbf{x}))$ *is $mz$-transportable in $\mathcal{D}$ if there exists $\mathbf{Z}'_i \subseteq \mathbf{Z}_i$ such that all paths from $\mathbf{Z}'_i$ to $\mathbf{Y}$ are blocked by $\mathbf{X}$, $(\mathbf{S_i} \perp\!\!\!\perp \mathbf{Y}|\mathbf{X}, \mathbf{Z}'_i)_{D^{(i)}_{\overline{\mathbf{X}, \mathbf{z}'_i}}}$, and $R$ is computable from $do(\mathbf{Z}_i)$.*

Remarkably, randomizing $Z_2$ when applying Corollary 1 was instrumental to yield transportability in the previous example, despite the fact that the directed paths from $Z_2$ to $Y$ were *not* blocked by $X$, which suggests how different this transportability is from $z$-identifiability. So, it is not immediately obvious how to combine the topological relations of $\mathbf{Z_i}$'s with $\mathbf{X}$ and $\mathbf{Y}$ in order to create a general condition for $mz$-transportability, the relationship between the distributions in the different domains can get relatively intricate, but we defer this discussion for now and consider a simpler case.

It is not usually trivial to pursue a derivation of $mz$-transportability in causal calculus, and next we show an example in which such derivation does not even exist. Consider again the diagrams in Fig. 2(a,b), and assume that randomized experiments are available over $\{Z_1\}$ in $\pi_a$ and $\{Z_2\}$ in $\pi_b$.

**Theorem 2.** $P^*(y|do(x))$ *is not $mz$-transportable in Fig. 2(a,b) with $\mathbf{Z_a} = \{Z_1\}$ and $\mathbf{Z_b} = \{Z_2\}$.*

*Proof.* Formally, we need to display two models $M_1, M_2$ such that the following relations hold (as implied by Def. 3):

$$
\begin{cases}
P^{(a)}_{M_1}(Z_1, X, Z_2, Y) = P^{(a)}_{M_2}(Z_1, X, Z_2, Y), \\
P^{(b)}_{M_1}(Z_1, X, Z_2, Y) = P^{(b)}_{M_2}(Z_1, X, Z_2, Y), \\
P^{(a)}_{M_1}(X, Z_2, Y|do(Z_1)) = P^{(a)}_{M_2}(X, Z_2, Y|do(Z_1)), \\
P^{(b)}_{M_1}(Z_1, X, Y|do(Z_2)) = P^{(b)}_{M_2}(Z_1, X, Y|do(Z_2)), \\
P^*_{M_1}(Z_1, X, Z_2, Y) = P^*_{M_2}(Z_1, X, Z_2, Y),
\end{cases}
\tag{13}
$$

for all values of $Z_1$, $X$, $Z_2$, and $Y$, and also,

$$
P^*_{M_1}(Y|do(X)) \neq P^*_{M_2}(Y|do(X)),
\tag{14}
$$

for some value of $X$ and $Y$.

Let $\mathbf{V}$ be the set of observable variables and $\mathbf{U}$ be the set of unobservable variables in $\mathcal{D}$. Let us assume that all variables in $\mathbf{U} \cup \mathbf{V}$ are binary. Let $U_1, U_2 \in \mathbf{U}$ be the common causes of $Z_1$ and $X$ and $Z_2$, respectively; let $U_3, U_4, U_5 \in \mathbf{U}$ be a random disturbance exclusive to $Z_1$, $Z_2$, and $Y$, respectively, and $U_6 \in \mathbf{U}$ be an extra random disturbance exclusive to $Z_2$, and $U_7, U_8 \in \mathbf{U}$ to $Y$. Let $S_a$ and $S_b$ index the model in the following way: the tuples $\langle S_a = 1, S_b = 0 \rangle$, $\langle S_a = 0, S_b = 1 \rangle$, $\langle S_a = 0, S_b = 0 \rangle$ represent domains $\pi_a$, $\pi_b$, and $\pi^*$, respectively. Define the two models as follows:

$$
M_1 = \begin{cases}
Z_1 = U_1 \oplus U_2 \oplus (U_3 \wedge S_a) \\
X = Z_1 \oplus U_1 \\
Z_2 = (X \oplus U_2 \oplus (U_4 \wedge S_a)) \vee \overline{U_6} \\
Y = (Z_2 \wedge U_5) \oplus (\overline{U_5} \wedge U_7) \oplus (S_b \wedge U_8)
\end{cases}
\qquad
M_2 = \begin{cases}
Z_1 = U_1 \oplus U_2 \oplus (U_3 \wedge S_a) \\
X = U_1 \\
Z_2 = (U_4 \wedge S_a \wedge U_6) \oplus \overline{U_6} \\
Y = (Z_2 \wedge U_5) \oplus (\overline{U_5} \wedge U_7) \oplus (S_b \wedge U_8)
\end{cases}
$$

where $\oplus$ represents the *exclusive or* function. Both models agree in respect to $P(\mathbf{U})$, which is defined as $P(U_i) = 1/2$, $i = 1, ..., 8$. It is not difficult to evaluate these models and note that the constraints given in Eqs. (13) and (14) are satisfied (including positivity), the theorem follows. $\quad\square$

## 4 Algorithm for computing $mz$-transportability

In this section, we build on previous analyses of identifiability [7, 21, 22, 23] in order to obtain a mechanic procedure in which a collection of selection diagrams and experimental data is inputted, and the procedure returns a transport formula whenever it is able to produce one. More specifically,

PROCEDURE $\mathbf{TR^{mz}}(\mathbf{y}, \mathbf{x}, \mathcal{P}, \mathcal{I}, \mathcal{S}, \mathcal{W}, D)$
INPUT: $\mathbf{x}, \mathbf{y}$: value assignments; $\mathcal{P}$: local distribution relative to domain $\mathcal{S}$ ($\mathcal{S} = 0$ indexes $\pi^*$) and active experiments $\mathcal{I}$; $\mathcal{W}$: weighting scheme; $D$: backbone of selection diagram; $\mathbf{S_i}$: selection nodes in $\pi_i$ ($\mathbf{S_0} = \emptyset$ relative to $\pi^*$); [The following set and distributions are globally defined: $\mathbf{Z}_i, P^*, P_{\mathbf{Z}_i}^{(i)}$.]

OUTPUT: $P_{\mathbf{x}}^*(\mathbf{y})$ in terms of $P^*, P_{\mathbf{Z}}^*, P_{\mathbf{Z}_i}^{(i)}$ or $FAIL(D, C_0)$.

1    **if** $\mathbf{x} = \emptyset$, **return** $\sum_{\mathbf{V} \setminus \mathbf{Y}} \mathcal{P}$.
2    **if** $\mathbf{V} \setminus An(\mathbf{Y})_D \neq \emptyset$, **return** $\mathbf{TR^{mz}}(\mathbf{y}, \mathbf{x} \cap An(\mathbf{Y})_D, \sum_{\mathbf{V} \setminus An(\mathbf{Y})_D} \mathcal{P}, \mathcal{I}, \mathcal{S}, \mathcal{W}, D_{An(\mathbf{Y})})$.
3    set $\mathbf{W} = (\mathbf{V} \setminus \mathbf{X}) \setminus An(\mathbf{Y})_{D_{\overline{\mathbf{X}}}}$.
      **if** $\mathbf{W} \neq \emptyset$, **return** $\mathbf{TR^{mz}}(\mathbf{y}, \mathbf{x} \cup \mathbf{w}, \mathcal{P}, \mathcal{I}, \mathcal{S}, \mathcal{W}, D)$.
4    **if** $\mathcal{C}(D \setminus \mathbf{X}) = \{C_0, C_1, ..., C_k\}$, **return** $\sum_{\mathbf{V} \setminus \{\mathbf{Y}, \mathbf{X}\}} \prod_i \mathbf{TR^{mz}}(c_i, \mathbf{v} \setminus c_i, \mathcal{P}, \mathcal{I}, \mathcal{S}, \mathcal{W}, D)$.
5    **if** $\mathcal{C}(D \setminus \mathbf{X}) = \{C_0\}$,
6      **if** $\mathcal{C}(D) \neq \{D\}$,
7        **if** $C_0 \in \mathcal{C}(D)$, **return** $\prod_{i|V_i \in C_0} \sum_{\mathbf{V} \setminus V_D^{(i)}} \mathcal{P} / \sum_{\mathbf{V} \setminus V_D^{(i-1)}} \mathcal{P}$.
8      **if** $(\exists C')C_0 \subset C' \in \mathcal{C}(D)$,
        **for** $\{i|V_i \in C'\}$, **set** $\kappa_{\mathbf{i}} = \kappa_i \cup v_D^{(i-1)} \setminus C'$.
        **return** $\mathbf{TR^{mz}}(\mathbf{y}, \mathbf{x} \cap C', \prod_{i|V_i \in C'} \mathcal{P}(V_i | V_D^{(i-1)} \cap C', \kappa_{\mathbf{i}}), \mathcal{I}, \mathcal{S}, \mathcal{W}, C')$.
9    **else**,
10     **if** $\mathcal{I} = \emptyset$, **for** $i = 0, ..., |\mathcal{D}|$,
        **if** $((\mathbf{S_i} \perp\!\!\!\perp \mathbf{Y} \mid \mathbf{X})_{D_{\overline{\mathbf{X}}}^{(i)}} \wedge (\mathbf{Z_i} \cap \mathbf{X} \neq \emptyset))$, $E_i = \mathbf{TR^{mz}}(\mathbf{y}, \mathbf{x} \setminus \mathbf{z_i}, \mathcal{P}, \mathbf{Z_i} \cap \mathbf{X}, i, \mathcal{W}, D \setminus \{\mathbf{Z_i} \cap \mathbf{X}\})$.
11    **if** $|\mathbf{E}| > 0$, **return** $\sum_{i=1}^{|\mathbf{E}|} w_i^{(j)} E_i$.
12    **else**, **FAIL**$(D, C_0)$.

Figure 3: Modified version of identification algorithm capable of recognizing $mz$-transportability.

our algorithm is called $\mathbf{TR^{mz}}$ (see Fig. 3), and is based on the $C$-component decomposition for identification of causal effects [22, 23] (and a version of the identification algorithm called **ID**).

The rationale behind $\mathbf{TR^{mz}}$ is to apply Tian's factorization and decompose the target relation into smaller, more manageable sub-expressions, and then try to evaluate whether each sub-expression can be computed in the target domain. Whenever this evaluation fails, $\mathbf{TR^{mz}}$ tries to use the experiments available from the target and, if possible, from the sources; this essentially implements the declarative condition delineated in Theorem 1. Next, we consider the soundness of the algorithm.

**Theorem 3** (soundness). *Whenever $\mathbf{TR^{mz}}$ returns an expression for $P_{\mathbf{x}}^*(\mathbf{y})$, it is correct.*

In the sequel, we demonstrate how the algorithm works through the $mz$-transportability of $Q = P^*(y|do(x))$ in Fig. 2(c,d) with $\mathbf{Z}^* = \{Z_1\}$, $\mathbf{Z_a} = \{Z_2\}$, and $\mathbf{Z_b} = \{Z_1\}$.

Since $(\mathbf{V} \setminus \mathbf{X}) \setminus An(\mathbf{Y})_{D_{\overline{\mathbf{X}}}} = \{Z_2\}$, $\mathbf{TR^{mz}}$ invokes line 3 with $\{Z_2\} \cup \{X\}$ as interventional set. The new call triggers line 4 and $\mathcal{C}(D \setminus \{X, Z_2\}) = \{C_0, C_1, C_2, C_3\}$, where $C_0 = D_{Z_1}$, $C_1 = D_{Z_3}$, $C_2 = D_U$, and $C_3 = D_{W,Y}$, we invoke line 4 and try to $mz$-transport individually $Q_0 = P_{x,z_2,z_3,u,w,y}^*(z_1)$, $Q_1 = P_{x,z_1,z_2,u,w,y}^*(z_3)$, $Q_2 = P_{x,z_1,z_2,z_3,w,y}^*(u)$, and $Q_3 = P_{x,z_1,z_2,z_3,u}^*(w,y)$. Thus the original problem reduces to try to evaluate the equivalent expression $\sum_{z_1,z_3,u,w} P_{x,z_2,z_3,u,w,y}^*(z_1) P_{x,z_1,z_2,u,w,y}^*(z_3) P_{x,z_1,z_2,z_3,w,y}^*(u) P_{x,z_1,z_2,z_3,u}^*(w,y)$.

First, $\mathbf{TR^{mz}}$ evaluates the expression $Q_0$ and triggers line 2, noting that all nodes can be ignored since they are not ancestors of $\{Z_1\}$, which implies after line 1 that $P_{x,z_2,z_3,u,w,y}^*(z_1) = P^*(z_1)$.

Second, $\mathbf{TR^{mz}}$ evaluates the expression $Q_1$ triggering line 2, which implies that $P_{x,z_1,z_2,u,w,y}^*(z_3) = P_{x,z_1,z_2}^*(z_3)$ with induced subgraph $D_1 = D_{X,Z_1,Z_2,Z_3}$. $\mathbf{TR^{mz}}$ goes to line 5, in which the local call $\mathcal{C}(D \setminus \{X, Z_1, Z_2\}) = \{D_{Z_3}\}$. Thus it proceeds to line 6 testing whether $\mathcal{C}(D \setminus \{X, Z_1, Z_2\})$ is different from $D_1$, which is false. In this call, ordinary identifiability would fail, but $\mathbf{TR^{mz}}$ proceeds to line 9. The goal of this line is to test whether some experiment can help for computing $Q_1$. In this case, $\pi_a$ fails immediately the test in line 10, but $\pi_b$ and $\pi^*$ succeed, which means experiments in these domains may eventually help; the new call is $P_{x,z_2}^{(i)}(z_3)_{D \setminus Z_1}$, for $i = \{b, *\}$ with induced graph $D_1' = D_{X,Z_2,Z_3}$. Finally, $\mathbf{TR^{mz}}$ triggers line 8 since $X$ is not part of $Z_3$'s components in $D_1'$ (or, $Z_3 \in C' = \{Z_2 \dashleftarrow\dashrightarrow Z_3\}$), so line 2 is triggered since $Z_2$ is no longer an ancestor of $Z_3$ in $D_1'$, and then line 1 is triggered since the interventional set is empty in this local call, so $P_{x,z_1,z_2}^*(z_3) = \sum_{Z_2'} P_{z_1}^{(i)}(z_3|x, Z_2') P_{z_1}^{(i)}(Z_2')$, for $i = \{b, *\}$.

Third, evaluating the expression $Q_2$, $\mathbf{TR^{mz}}$ goes to line 2, which implies that $P^*_{x,z_1,z_2,z_3,w,y}(u) = P^*_{x,z_1,z_2,z_3,w}(u)$ with induced subgraph $D_2 = D_{X,Z_1,Z_2,Z_3,W,U}$. $\mathbf{TR^{mz}}$ goes to line 5, and in this local call $\mathcal{C}(D \setminus \{X, Z_1, Z_2, Z_3, W\}) = \{D_U\}$, and the test in 6 succeed, since there are more components in $D$. So, it triggers line 8 since $W$ is not part of $U$'s component in $D_2$. The algorithm makes $P^*_{x,z_1,z_2,z_3,w}(u) = P^*_{x,z_1,z_2,z_3}(u)_{D_2|W}$ (and update the working distribution); note that in this call, ordinary identifiability would fail since the nodes are in the same $C$-component and the test in line 6 fails. But $\mathbf{TR^{mz}}$ proceeds to line 9 trying to find experiments that can help in $Q_2$'s computation. In this case, $\pi_b$ cannot help but $\pi_a$ and $\pi^*$ perhaps can, noting that new calls are launched for computing $P^{(a)}_{x,z_1,z_3}(u)_{D_2 \setminus Z_2|W}$ relative to $\pi_a$, and $P^*_{x,z_2,z_3}(u)_{D_2 \setminus Z_1|W}$ relative to $\pi^*$ with the corresponding data structures set. In $\pi_a$, the algorithm triggers line 7, which yields $P^{(a)}_{x,z_1,z_3}(u)_{D_2 \setminus Z_2|W} = P^{(a)}_{z_2}(u|w, z_3, x, z_1)$, and a bit more involved analysis for $\pi_b$ yields (after simplification) $P^*_{x,z_2,z_3}(u)_{D_2 \setminus Z_1|W} = \left( \sum_{Z'_2} P^*_{z_1}(u|w, z_3, x, Z'_2) P^*_{z_1}(z_3|x, Z'_2) P^*_{z_1}(Z'_2) \right) / \left( \sum_{Z''_2} P^*_{z_1}(z_3|x, Z''_2) P^*_{z_1}(Z''_2) \right)$.

Fourth, $\mathbf{TR^{mz}}$ evaluates the expression $Q_3$ and triggers line 5, $\mathcal{C}(D \setminus \{X, Z_1, Z_2, Z_3, U\}) = D_{W,Y}$. In turn, both tests at lines 6 and 7 succeed, which makes the procedure to return $P^*_{x,z_1,z_2,z_3,u}(w, y) = P^*(w|z_3, x, z_1, z_2) P^*(y|w, x, z_1, z_2, z_3, u)$.

The composition of the return of these calls generates the following expression:

$$
\begin{aligned}
P^*_x(y) &= \sum_{z_1, z_3, w, u} P^*(z_1) \left( w_1^{(1)} \sum_{Z'_2} P^*_{z_1}(z_3|x, Z'_2) P^*_{z_1}(Z'_2) + w_2^{(1)} \sum_{Z'_2} P^{(b)}_{z_1}(z_3|x, Z'_2) P^{(b)}_{z_1}(Z'_2) \right) \\
&\quad \left( w_1^{(2)} \left( \sum_{Z'_2} P^*_{z_1}(u|w, z_3, x, Z'_2) P^*_{z_1}(z_3|x, Z'_2) P^*_{z_1}(Z'_2) \right) / \left( \sum_{Z''_2} P^*_{z_1}(z_3|x, Z''_2) P^*_{z_1}(Z''_2) \right) \right. \\
&\quad \left. + w_2^{(2)} P^{(a)}_{z_2}(u|w, z_3, x, z_1) \right) P^*(w|x, z_1, z_2, z_3) \, P^*(y|x, z_1, z_2, z_3, w, u)
\end{aligned}
\tag{15}
$$

where $w_i^{(k)}$ represents the weight for each factor in estimand $k$ ($i = 1, ..., n_k$), and $n_k$ is the number of feasible estimands of $k$. Eq. (15) depicts a powerful way to estimate $P^*(y|do(x))$ in the target domain, and depending on weighting choice a different estimand will be entailed. For instance, one might use an analogous to *inverse-variance weighting*, which sets the weights for the normalized inverse of their variances (*i.e.*, $w_i^{(k)} = \sigma_i^{-2} / \sum_{j=1}^{n_k} \sigma_j^{-2}$, where $\sigma_j^2$ is the variance of the $j$th component of estimand $k$). Our strategy resembles the approach taken in meta-analysis [4], albeit the latter usually disregards the intricacies of the relationships between variables, so producing a statistically less powerful estimand. Our method leverages this non-trivial and highly structured relationships, as exemplified in Eq. (15), which yields an estimand with less variance and statistically more powerful.

## 5 Conclusions

In this paper, we treat a special type of transportability in which experiments can be conducted only over limited sets of variables in the sources and target domains, and the goal is to infer whether a certain effect can be estimated in the target using the information scattered throughout the domains. We provide a general sufficient graphical conditions for transportability based on the causal calculus along with a necessary condition for a specific scenario, which should be generalized for arbitrary structures. We further provide a procedure for computing transportability, that is, generate a formula for fusing the available observational and experimental data to synthesize an estimate of the desired causal effects. Our algorithm also allows for generic weighting schemes, which generalizes standard statistical procedures and leads to the construction of statistically more powerful estimands.

**Acknowledgment**

The work of Judea Pearl and Elias Bareinboim was supported in part by grants from NSF (IIS-1249822, IIS-1302448), and ONR (N00014-13-1-0153, N00014-10-1-0933). The work of Sanghack Lee and Vasant Honavar was partially completed while they were with the Department of Computer Science at Iowa State University. The work of Vasant Honavar while working at the National Science Foundation (NSF) was supported by the NSF. The work of Sanghack Lee was supported in part by the grant from NSF (IIS-0711356). Any opinions, findings, and conclusions contained in this article are those of the authors and do not necessarily reflect the views of the sponsors.

## Footnotes

[1]The machine learning literature has been concerned about discrepancies among domains in the context, almost exclusively, on predictive or classification tasks as opposed to learning causal or counterfactual measures [16, 17]. Interestingly enough, recent work on anticausal learning moves towards more general modalities of learning and also leverages knowledge about the underlying data-generating structure [18, 19].

[2]We will use $P_{\mathbf{x}}(\mathbf{y} \mid \mathbf{z})$ interchangeably with $P(\mathbf{y} \mid do(\mathbf{x}), \mathbf{z})$.

[3]We use the structural interpretation of causal diagrams as described in [9, pp. 205].

[4]A typical example is whether we can estimate the effect of cholesterol ($X$) on heart failure ($Y$) by experiments on diet ($Z_1$) given that cholesterol levels cannot be randomized [20].

[5]The results presented here are also valid in other formalisms for causality based on potential outcomes.

[6]As discussed in the reference, the assumption of no structural changes between domains can be relaxed, but some structural assumptions regarding the discrepancies between domains must still hold.

[7]Transportability assumes that enough structural knowledge about both domains is known in order to substantiate the production of their respective causal diagrams. In the absence of such knowledge, *causal discovery* algorithms might be used to infer the diagrams from data [8, 9].

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
