[Reviews · NeurIPS 2013]

Submitted by Assigned_Reviewer_1

Previously it has been shown that do-calculus is a sound inferential machinery for estimating a causal effect from a causal diagram and a set of observations and interventions. This paper further proves that it is not only sound, but also complete, meaning that every valid equality between probabilities defined on a semi-Markovian graph can be obtained through finite applications of the three rules of do-calculus. Moreover, the paper studies mz-transportability, which unifies those previously studied special cases of meta-identifiability. The authors proposed a complete algorithm to determine if a causal effect is mz-transportable, and if it is, outputs a transport formula for estimating the causal effect.

This paper is well written and made a clear contribution to identification and meta-transportability of causal effects. I enjoyed reading the paper, and have some comments regarding the presentation. A weakness is that the relationship between the present work and previous work is not clearly discussed. For instance, in Section 3 , it would be very helpful if the authors could make it explicit how their completeness result of do-calculus is more general than that is given in [16]. Furthermore, it is inevitable that the paper gives a number of definitions, but for better readability to people from other fields, it would be appreciated if the authors could give the intuition behind certain concepts (such as hedge).

Minor points:
1. Line 145: "(the) minimum observations of p and q."
2. Line 161: "(for) P(y=1|do(y=0)))."
3. Line 208: "abd (the) rest."
4. Line 216: "A domain-specifc selection variables."
5. Line 247: "conditioning variable in do-terms are."
6. Line 373: Here it is the first time that the authors have used mzTR; please refer to Figure 2.
Summary: A nice and solid paper. I enjoying reading it.

Submitted by Assigned_Reviewer_2

A semi-Markovian graph induces equivalence classes of probabilistic expressions that hold for every distribution that respects the graph structure.
(for example [p(y|do(x))] = [p(y)] for every distribution p on a graph G: X->Y)
In the first part of the paper, the authors prove that equality of equivalence classes [p] = [q] can be determined by applying three manipulation rules (the do-calculus) a finite number of times. (hence, the do-calculus is complete)
In the second part they use their result to design an algorithm that decides the transportability of results across multiple domains. By this, it is meant that the algorithm decides, whether an interventional probability can be obtained from studies that have taken place under slighty different conditions (e.g. varying exogeneous variables, varying structural functions). Based on previous work, this variation of conditions can be formalized through so called selection diagrams. Since these are semi-Markoviaan graphs the result of the first part of the paper can be used to prove completeness of the proposed algorithm that is based on the do-calculus.


I am not able give guarantees on the mathematical correctness of the proofs.


Significance
The results of the paper are significant since the completeness of the do-calculus proves the completeness of algorithms based on the three manipulation rules.
Also the author's definition of mz-transportability subsumes a number of different notions of transportability and is decidable through the presented algorithm.

Originality
Completeness of the do-calculus for semi-markovian models has only been known for the problem of identification of expressions p(y|do(x)) and not for general probabilistic expressions, like p(y|do(x),z).

Clarity:
In my opinion the paper is written for a very specialized audience. The style is formal, it is highly incremental, not self-contained (referencing the literature, instead of repeating/motivating definitions, maybe partly do to space constraints). The results could be made accessible to a wider audience (parts of the introduction are redundant)
Summary: Results of the paper are important (completeness of do-calculus for semi-Markovian models and a new notion of transportability together with an algorithm to decide it)
However, the presentation is very dense, depends on previous work (not self-contained) and could be made accessible to a wider audience.

Submitted by Assigned_Reviewer_5

This paper showed the completeness of do-calculus for nonparametric estimation of a causal effect in a semi-Markovian graph structured causal diagram to which an arbitrary set of interventional and observation distributions conforms. It introduced a problem to infer a causal effect of treatment variables on observables in a target domain by combining data from experiments on simultaneously controllable subsets of variables from multiple domains and formulated it as mz-transportability. It further presented a complete algorithm for determining causal effects.

Quality of the paper
This paper showed the completeness of do-calculus in a semi-Markovian causal graph in a very generic setting. This consequence clarifies the conditions for the valid causal inference and ensures the validity of the causal inference under the condition. It further formulate a measure named mz-transportability and an algorithm to provide a transport formula for estimating the causal effect. These consequences provides a firm basis of the causal inference. In this regard, the work presented in this paper is highly meaningful.
Mathematical formulations and their associated proofs seem to be valid, and thus their technical quality is considered to be high.
However, the contents of the paper seem to be too much to fill in the limited length. We could not follow the explanations and the mathematical derivations perfectly since there are some significant explanation gaps because of the limited space for the explanation. For example, at the beginning of the subsection 4.1, the authors claim that "We use the notational conventions and definitions for causal models introduced in [2, 15, 12]." In this sense, this paper is not self-contained. In addition, in the latter half of the subsection 4.2, the authors missed to explain the definition of "c-componets". Moreover, the entire description of the paper lacks the explanation through some comprehensible examples. The use of the examples eases a lot the comprehension of the readers particularly in this type of theoretical papers. Though these are issues of the paper presentation, this affect the value of the paper.

Clarity
The explanation and the proofs provided in the paper seems mathematically rigorous. However, as pointed out above, the contents of the paper is too much for the NIPS page limitation. The characterization of the mz-transportation can be far more reduced to provide some basis only for the algorithm in the subsection 4.3. The characterization can be presented in its journal paper version. By presenting the essential skeleton of the contents, the authors can show sufficient value of this study.

Originality
The mathematical and technical quality of this paper seems high. However, the scope of the paper is rather incremental, and it does not provide entirely novel principles in this research field.

Significance
The consequence of this paper widely enables causal analysis over multiple domains while ensuring the identifiability of the causal structures. The paper is significant in this regard. But, the paper does not include any experimental validation of the proposed consequences. If some numerical experiments applying the proposed algorithms were demonstrated, the significance of the result could be more clearly indicated.
Summary: The technical consequence provided in this paper is rigorous and potentially significant for the wide applicability of causal inference. However, the paper does not provide sufficient explanations of mathematical definitions and comprehensive examples. It further failed to demonstrate the validation of the results through comprehensive examples including some numerical experiments.
Author Feedback

Author rebuttal: The paper was changed including the addition of some authors. We tried to adhere to the reviewers suggestions as much as possible, but the structure of the paper changed, significant parts were removed and others were added to the manuscript.